# Transplant Eligible and Ineligible Elderly Patients with AML—A Genomic Approach and Next Generation Questions

**DOI:** 10.3390/biomedicines12050975

**Published:** 2024-04-29

**Authors:** Paul Sackstein, Alexis Williams, Rachel Zemel, Jennifer A. Marks, Anne S. Renteria, Gustavo Rivero

**Affiliations:** 1Lombardi Cancer Institute, School of Medicine, Georgetown University, Washington, DC 20007, USA; paul.e.sackstein@medstar.net (P.S.); rachel.a.zemel@medstar.net (R.Z.); jennifer.a.marks@medstar.net (J.A.M.); 2Department of Medicine, New York University, New York, NY 10016, USA; alexis.williams@nyulangone.org

**Keywords:** elderly, acute myelogenous leukemia, pathogenic variants, remission induction, allogeneic HCT, relapse

## Abstract

The management of elderly patients diagnosed with acute myelogenous leukemia (AML) is complicated by high relapse risk and comorbidities that often preclude access to allogeneic hematopoietic cellular transplantation (allo-HCT). In recent years, fast-paced FDA drug approval has reshaped the therapeutic landscape, with modest, albeit promising improvement in survival. Still, AML outcomes in elderly patients remain unacceptably unfavorable highlighting the need for better understanding of disease biology and tailored strategies. In this review, we discuss recent modifications suggested by European Leukemia Network 2022 (ELN-2022) risk stratification and review recent aging cell biology advances with the discussion of four AML cases. While an older age, >60 years, does not constitute an absolute contraindication for allo-HCT, the careful patient selection based on a detailed and multidisciplinary risk stratification cannot be overemphasized.

## 1. Introduction

With advancing age, the incidence of acute myeloid leukemia (AML) increases, and the disease brings a more adverse prognosis [1,2,3], with a median age at diagnosis of 68 years [4]. Typically, above the age of 60 years, AML is characterized by biologically adverse mutations often associated with chemoresistance, patients’ health comorbidities that impact performance status [5], and socioeconomic factors impairing access to appropriate treatment and support [6]. We are witnessing a therapeutic momentum supported by the recent FDA approvals of several AML targeted agents, although vulnerable older patients continue to experience unacceptable poor outcomes. For patients with non-favorable risk AML, once a complete remission (CR) is achieved, most treating physicians rely on allo-HCT as a potentially curative intervention. However, transplant eligibility weighs the risks of transplant-related mortality (TRM) against the potential benefits of improved survival, including a cure [6]. With the development of reduced-intensity conditioning (RIC) regimens, AML management in older patients with reduced fitness do not absolutely preclude allo-HCT [7]. Allo-HCT may be considered in elderly fit patients with intermediate to high-risk AML, who would otherwise experience an inferior AML-specific survival, when compared to younger patients [8]. In this review, we address challenging aspects related to aging and predisposition to hematopoietic malignancies, the recent modifications proposed by the ELN-2022 risk stratification [9], and present four clinical cases to discuss the management of elderly patients diagnosed with favorable, intermediate, and adverse ELN-2022 risk AML and their indication and eligibility to allo-HCT consolidation.

## 2. The Biology of Aging, Leukemia, and Bone Marrow Transplantation

Inevitably, aging is associated with an increased prevalence of clonal hematopoiesis of indeterminate potential (CHIP), that is linked to an increased risk for developing hematopoietic malignancies and cardiovascular diseases [10,11]. The sequence of events leading to overt myeloid malignancies is not entirely understood. Clonal dominance is often detected, and expansion of oligoclonal hematopoiesis develops over time. Additional leukemogenic mechanisms, including epigenetic and transcriptomic deregulation, are likely to contribute to AML development.

Next-generation sequencing (NGS) has identified age-related clonal hematopoietic mutations such as *DNMT3A*, *ASXL1,* and *TET2* in elderly patients with AML [1]. While these mutations are widely found in myelodysplastic syndrome (MDS) and myeloproliferative neoplasms (MPN), the high incidence of these pathogenic variants in elderly patients with AML suggests that their diseases retain ontogeny and direct correlation with a pre-existing MDS or MPN.

The poorer outcomes associated with AML diagnosed in patients ≥ 60 years has been attributed to increased prevalence of monosomal and complex karyotypic abnormalities, medical comorbidities, and inability to tolerate intensive induction chemotherapy [12,13,14,15].

Previously, an age of >55 years and diffusion capacity of the lung carbon monoxide (DLCO) ≤ 60% were considered contraindications for allo-HCT [16,17]. However, RIC for these patients can have acceptable outcomes [17], provided they are in complete remission (CR), and have left ventricular ejection fraction (LVEF) ≥ 40% [18]. Given the high risk for post-allo-HCT fatal outcome, scoring systems incorporate multiple patients and donors and related factors have been developed and validated [19,20,21]. In addition, previous studies showed similar rates of 2-year overall survival (OS) for younger and older patients ≥ 65 years with AML undergoing allo-HCT [22], supporting that advanced chronologic age alone should not represent a barrier to transplant. Physiologic age, which varies among patients with the same chronological age, and the incorporation of medical comorbidities, frailty, and functional status, may better predict tolerance of intensive therapy and transplant eligibility [13].

## 3. European Leukemia Network-2022 Stratification Refinement

Since 2017, sequential updates in AML-related data have revised classification risks, emphasizing the need to adjust risk assessment criteria [9]. In addition to baseline genetic characterization, evaluating response to initial therapy and early measurable residual disease (MRD) have emerged as crucial to a patient’s disease risk assignment [23]. Notably, patients previously categorized as having favorable-risk AML might have been reclassified as having intermediate-risk disease and vice versa, based on the presence or absence of MRD, particularly relevant in cases of *NPM1*-mutant AML [24,25,26]. Key modifications to previous risk classifications include several significant changes. The *FLT3-ITD* allelic ratio no longer influences risk classification; instead, AML with *FLT3-ITD* (without adverse-risk genetics) is now categorized as intermediate-risk due to assay standardization challenges and the increasing significance of MRD [27]. AML with myelodysplasia-related gene mutations is classified as adverse risk, encompassing mutations in various genes beyond *ASXL1* and *RUNX1* [3,28,29,30,31]. Additionally, adverse-risk cytogenetic abnormalities in *NPM1*-mutated AML define adverse risk, based on poor outcomes observed in meta-analyses. Favorable-risk status now includes in-frame mutations affecting *CEBPA’*s basic leucine zipper region, regardless of their biallelic or monoallelic occurrence [32,33,34]. Furthermore, disease-defining recurring cytogenetic abnormalities are included in the adverse-risk group [35]. Hyperdiploid karyotypes with multiple trisomies are no longer considered complex karyotypes or of adverse risk [36]. These adjustments reflect the evolving understanding of AML pathogenesis and emphasize the importance of MRD assessment in refining treatment strategies and risk management protocols.

**CASE 1**—A 62-year-old male with a past medical history (PMH) of Type II diabetes mellitus who presented with a white blood cell (WBC) count of 68,000/µL. Initial flow cytometry showed an expanded monocytic-like population, CD34(−), CD38(+), HLADR-CD33(+) bright, and CD117(+).

### 3.1. Case 1 Overview, Favorable Risk ELN-2022

A key decision point is whether to pursue full-intensity “7 + 3” induction therapy or to use a reduced-intensity regimen such as azacytidine, a hypomethylating agent (HMA), and venetoclax, a BCL-2 inhibitor [37]. To decide, a therapy-related mortality (TRM) scoring is recommended and can predict if the risk related to a high-intensity induction would outweigh its benefits [38]. This patient’s TRM score, based on age, platelet count, and functional status, was in the low-risk category, with a predicted TRM of 2%–10% with high-intensity induction; he underwent “7 + 3”. The patient’s disease NGS showed *NPM1* p.W288, with a variant allele frequency (VAF) of 43.8%, and *IDH1* p.R132 mutations. Gemtuzumab ozogamicin (GO) (Figure 1) was added with the intention of optimizing induction response [39]. The Acute Leukemia French Association (ALFA) 0701 study group demonstrated that hyper-fractionated GO (lower antibody dose at 3 mg/m^2^ on days 1, 4, and 7) concurrently administered with intravenous cytarabine plus anthracycline was safe and efficacious in the therapy of AML [40]. Indeed, a decrease in MRD can be obtained with the anti-CD33 monoclonal antibody GO in *NMP1*-mutated AML patients [41] and other favorable ELN-2022 subgroups including core binding factor (CBF) AML [42], especially if administered in two sequential doses.

### 3.2. Recent Advances

#### Where Are We with Hypomethylating Agents Plus IDH1 or IDH2 Combinations?

Approximately 20% of AML carry an IDH1 or IDH2 mutation [43], and ivosidenib and enasidenib (Figure 1) have been approved for their treatment, respectively. Initial studies evaluated the efficacy of these agents in patients with relapsed/refractory (R/R) disease as well as newly diagnosed IDH1/2-mutated AML. In R/R AML, ivosidenib and enasidenib lead to an overall response rate (ORR) of 41.6% and 40.8%, respectively [44,45]. In newly diagnosed IDH1/2-mutated AML not eligible for standard induction, ivosidenib and enasidenib had a composite remission rate of 70.6% and 18%, respectively [46,47]. Favorable outcomes were also seen in combination with standard high-intensity induction chemotherapy such as “7 + 3” [48] as well.

The impact of HMA, such as azacytidine, with and without venetoclax for management of IDH1/2-mutated AML, is the subject of ongoing studies. In preclinical data, Chan et al. found that IDH1/2-mutated cancer cells have increased dependence on the anti-apoptotic activity of BCL-2 [43]. Pollyea et al. analyzed the IDH1/2-mutated subgroup of the phase III and phase 1b trials of venetoclax efficacy [49]. Venetoclax combined with azacytidine induced favorable outcomes when related to remission rates, OS, and duration of response compared to either treatment group in IDH1/2 wild-type patients or compared to mutated IDH1/2 patients treated with azacytidine alone. They also found that patients with IDH1/2 mutations and otherwise high-risk cytogenetics maintained their response to azacytidine/venetoclax.

In a phase 3 trial of induction-ineligible AML with IDH 1 mutations, Montesinos et al. studied ivosidenib combined with azacytidine compared to azacytidine alone and found an increased OS (24.0 vs. 7.9 months, *p* = 0.001) in the ivosidenib group [50]. A phase 1b trial investigated the triplet therapy of ivosidenib, venetoclax, and azacytidine; increased efficacy was found when compared to ivosidenib with venetoclax alone, as well as increased MRD-negative remissions [51]. For *IDH2*-mutated AML, a phase II trial of azacytidine and enasidenib observed a CR rate of 100% in induction-ineligible patients, and 58% in R/R patients [52].

**CASE 2**—A 64-year-old male with mitral valve prolapse status post repair and permanent pacemaker placement presented with renal failure in the setting of tumor lysis syndrome (TLS) and a WBC count of 57,000/µL. Bone marrow biopsy (BMBx) identified 90% blasts with monocytic differentiation and NGS showed t(9;11) with *KMT2A-MLLT3* fusion, and *FLT3-TKD*, *NRAS* (G12D), *NRAS* (G12S), *NRAS* (G13D), and *KRAS* (G12D) mutations.

### 3.3. Case 2 Overview (Intermediate Risk ELN-2022)

He received rasburicase and intravenous fluids leading to renal function recovery. After intensive induction with “7 + 3” (cytarabine and idarubicin) and midostaurin, he reached a CR with negative MRD by flow cytometry. The t(9;11) is associated, as per ELN-2022, with an intermediate risk prognosis and an overall 5-year OS of 30%. However, a more precise survival prediction can be obtained by integrating clinical (i.e., age and sex), laboratory (i.e., LDH, WBC, platelet count, peripheral blood, and marrow blast), karyotypic, and mutational analysis data to the risk calculation https://cancer.sanger.ac.uk/aml-multistage/ (accessed 12 January 2024) [53]; his predicted OS at 60 months is actually 40%. He is intended to undergo allo-HCT with an HLA 10/10 matched unrelated donor (MUD) after reduced intensity conditioning (RIC).

### 3.4. Recent Advances

#### What Are the New FLT3 Inhibitor Alternatives?

First generation FLT3 inhibitors such as sorafenib and midostaurin are multikinase inhibitor agents that also inhibit vascular endothelial growth factor (VEGF) and platelet derived growth factor (PDGF), two off-target related toxicities [54]. Consequently, there is an interest in developing novel selective FLT3 inhibitors bringing fewer adverse effects. Quizartinib, a selective second-generation type 2 FLT3 inhibitor, preferentially targets FLT3-ITD mutations. The Quantum FIRST trial randomized 539 young patients, median age of 58 years, with newly diagnosed FLT3-mutated AML to receive “7 + 3” induction chemotherapy followed by high-dose cytarabine (HiDAC) consolidation or consolidative allo-HCT with quizartinib or placebo [55]. Patients randomized to the quizartinib arm continued quizartinib maintenance for 3 years. The trial showed a significant improvement in median OS for patients receiving quizartinib compared to placebo (31.9 vs. 15.1 months, *p* = 0.032). Neither the Quantum FIRST trial nor the NCRI AML 18 trial [56] demonstrated a survival benefit to intensive induction chemotherapy with the addition of quizartinib in patients > 60 years with FLT3-mutated AML (Figure 1). The high TRM observed during the first 3 months of treatment is likely to have offset a survival benefit from quizartinib, with a 17.4% TRM rate in elderly patients [57]. Future studies with quizartinib should address these safety concerns before it can be routinely offered to elderly patients with AML.

Crenolanib, a novel type 1 FLT3 inhibitor, has specificity for both FLT3-ITD and FLT3-TKD mutations. In a phase II study, median OS was not reached at a 45-month follow-up in patients treated with crenolanib in combination with “7 + 3” induction, followed by HiDAC and/or allo-HCT, and crenolanib maintenance for 1 year [58]. A subgroup analysis of patients aged >60 years showed the safety and efficacy of combining crenolanib with “7 + 3” induction, consolidation, and maintenance (median OS of 20.2 months) [59]. There are currently no active trials of crenolanib in elderly patients with FLT3-mutated AML.

The LACEWING study, which evaluated gilteritinib in combination with azacitidine versus azacitidine alone in patients ≥ 65 years of age, showed higher rates of CR with the addition of gilteritinib (58.1% vs. 26.5%, *p* < 0.001) but no OS benefit; the study was terminated due to futility [60].

### 3.5. Triple Therapy for FLT3-ITD in Elderly Patients with AML

FLT3 mutations are seen in 30–40% of patients with AML, the majority being FLT3-ITD (20–25%), and a smaller proportion (5–10%) being FLT3-TKD mutations [2,3]. Although the proportion of FLT3-ITD mutations is lower in elderly patients with AML (18%), FLT3-ITD mutations are frequently identified in older individuals owing to increased rates of AML in the aging population [61]. A subgroup analysis of the VIALE-A trial showed inferior OS for patients with FLT3-ITD mutated AML receiving HMA plus venetoclax compared to FLT3 wild-type AML (median OS 9.9 vs. 14.7 months, respectively) [62]: FLT3-ITD mutations promote venetoclax resistance via BCL-2 and MCL-1 overexpression [63]. In vitro, the combination of gilteritinib with venetoclax decreased MCL-1 expression via proteasomal degradation, and restored venetoclax sensitivity, resulting in a synergistic effect [63]. Interestingly, synergy was not observed with other FLT3 inhibitors such as sorafenib, midostaurin, or quizartinib. These data offer a biological rationale for studied “triplet” therapy of gilteritinib, HMA, and venetoclax in FLT3-mutated AML [64].

FLT3-ITD mutated AML remains, however, an important unmet need in elderly patients. The RATIFY trial showed an impressive survival benefit with the addition of midostaurin to “7 + 3” induction or HMA plus venetoclax, compared to standard chemotherapy alone in the frontline setting (74.7 vs. 25.6 months, *p* = 0.009) [65]. However, this trial was restricted to patients ≤ 60 years, and the impact of such “triplet” therapies for elderly patients remains unclear. Retrospective series have suggested that triple therapies increase median OS compared to HMA plus venetoclax among elderly patients with FLT3-mutated AML [66]. In addition, data from a small prospective clinical trial enrolling 40 elderly patients with R/R and newly diagnosed FLT3-mutated AML showed high survival rates with azacitidine, venetoclax, and gilteritinib [67], suggesting a survival benefit and safe profile. CR rates were 95% among 20 patients with untreated disease with an estimated 1-year OS of 80%. Within the R/R cohort, the ORR was 74% (CR rate 21% with 43% MRD negativity by flow cytometry), however the 1-year OS was only 27%. No deaths were reported within 60 days of treatment using a reduced duration of venetoclax (7 days) in consolidation.

**CASE 3**—A 62-year-old male presented with progressive bicytopenia and circulating blasts on peripheral blood smear. BMBx revealed a t(11;17) by fluorescence in situ hybridization (FISH), and NGS showed *BCOR* (L1646T), *WT1* (G187R), and *WT1* (L408T) mutations with VAF 2.18%, 27.2%, and 0.6%, respectively.

### 3.6. Case 3 Overview (Adverse Risk ELN-2022)

Mixed lineage leukemia (MLL) spanning KM2TA abnormalities, and newly incorporated ELN-2022 AML with myelodysplasia-related changes (AML-MRC) such as *BCOR*, constitutes an adverse risk AML. He underwent induction with “7 + 3” (cytarabine and idarubicin). His response assessment BMBx on day 28 showed R/R disease with 40% blasts. HiDAC with 2000 mg/m^2^ was administered as salvage therapy; however, 17% residual marrow blasts persisted. He subsequently transitioned to HMA plus venetoclax and had detectable disease (6% blasts) after two consecutive cycles, carrying KM2TA abnormality. The patient was enrolled on a single agent menin inhibitor clinical trial. In view of his high-risk disease, as attested by his clinical course, he would be given the offer to proceed to allo-HCT, provided he attains a CR and has a good performance status and manageable comorbidities.

#### What Is the Role of Pre-Transplant Flow Cytometry Measurable Residual Disease (MRD)?

Efforts in flow cytometry methodology for validated MRD determinations are becoming progressively adopted. Typically, MRD is detected by inspection of “differential expression from normal” in subgroups of cells demonstrating more than one antigen abnormality from accepted expression at a precise stage of differentiation. An abnormal cell population is compared against normal and regenerative marrow [68]. The ELN-2022 risk stratification for diagnosis and management highlights that MRD informs: (a) surrogate for quality of remission status; (b) relapse risk estimation; and (c) imminent disease recurrence to allow preemptive intervention [9]. Leukemia associated immunophenotype (LAIP) can discriminate neoplastic cells from “normal stem cells/progenitors expression” to diagnose and monitor recurring clones after AML-directed therapy. The presence of MRD prior to allo-HCT is an adverse predictor of outcome [69], and in patients with favorable-risk disease, allo-HCT is recommended when there is detectable MRD after AML therapy.

### 3.7. Recent Advances

#### Menin Inhibitors

Menin inhibitors offer a promising approach for AML with KMT2A fusion proteins and NPM1 mutations. The interaction between menin and lysine methyltransferase 2A (*KMT2A*) plays a crucial role in AML pathogenesis, affected by *KMT2A* or Nucleoporin 98 (*NUP98*) gene rearrangements or Nucleophosmin 1 gene mutations (*NPM1* mt). Revumenib, formerly SNDX-5613, showed safety and clinical efficacy in refractory acute leukemias, particularly in *KMT2A* and *NPM1* mt AML, which are susceptible to apoptosis induction through BCL-2 inhibition, with added synergistic effects of menin inhibition [70,71].

A phase I/II trial investigated the all-oral combination of revumenib, venetoclax, and HMA ASTX727 in children and adults with R/R AML [72]. By 20 July 2023, eight patients with a median age of 27 years were enrolled. Most had undergone multiple lines of therapy, including venetoclax, HMA, and allo-HCT. The most common treatment-related adverse events included febrile neutropenia, hyperphosphatemia, nausea, and AST/ALT elevation. Encouragingly, all seven evaluable patients achieved morphologic remission, resulting in an ORR of 100%. Responses ranged from complete remission (CR) to partial response (PR), with some patients transitioning to allo-HCT following treatment response. Despite one dose-limiting toxicity (DLT) and manageable adverse events, the combination therapy demonstrated both acceptable safety and high efficacy in R/R myeloid leukemias with *KMT2A*, *NPM1*, or *NUP98* mutations. Additionally, JNJ-75276617, targeting the menin–KMT2A interaction, exhibited promising preclinical activity in *KMT2A*-rearranged or *NPM1*-mutated leukemias [73]. The ongoing Phase 1 trial (NCT04811560) enrolled 58 patients by 8 April 2023, most with R/R AML and a median age of 63 years [74]. DLTs were observed in 9% of patients, but significant reductions in bone marrow disease burden were also noted, especially at higher dose levels, with an ORR of 50%. Preliminary pharmacodynamic data indicated biologic activity among responders.

Both revumenib and JNJ-75276617 show promise in treating R/R AML with *KMT2A* or *NPM1* alterations. Ongoing trials aim to determine optimal dosing and validate their efficacy in this challenging patient population, supporting continued investigation into these combination therapies.

## 4. Transplant Considerations

### 4.1. Transplant Eligible Elderly Patients with AML

The median survival of patients ≥ 60 years diagnosed with AML ranges from 6 to 10 months, and remission rates and OS at 1 year are 40% and 15%, respectively [75]. Prior to the year 2000, improvements in survival were mostly achieved in patients < 60 years through dose intensification of conventional chemotherapy and use of allo-HCT. However, the outcomes for patients ≥ 60 years remained dismal overall [76]. With the emergence of more tolerable and effective induction regimens for older patients with AML, the risk-adapted recommendations for allo-HCT consolidation therapy started to be revisited in light of several promising new drugs.

Allo-HCT is a curative treatment for several hematological diseases through the graft-versus-leukemia (GVL) effect, and has the potential to offer longer term disease-free survival (DFS) [7]. Traditionally, myeloablative chemotherapy conditioning (MAC) regimens were the standard prior to donor stem cell infusion but were administered to patients up to age 50 due to concerns related to their tolerability in patients with significant comorbidities. RIC regimens granted access to allo-HCT to older patients, especially if in first CR and with minimal comorbidities, offering improved outcomes including DFS and acceptable non-relapse mortality (NRM) rates, becoming a viable option as consolidation therapy. The aspect of being in first CR is of significant importance as RIC is associated with higher risk for disease relapse when compared to MAC.

While RIC and non-myeloablative conditioning (NMA) regimens have allowed older patients to undergo a more tolerable allo-HCT, these conditioning regimens still carry a significant risk of NRM as well as higher relapse rates. When assessing a potential patient for allo-HCT, the entire allo-HCT procedure needs to be considered as follows: the induction chemotherapy used, the degree of myeloablation to be achieved by the RIC or NMA regimen prior to the donor stem cell infusion, the expected disease relapse risk, and the option of a maintenance therapy post-allo-HCT. Current NRM rates (25% to 35%) [6] and cumulative incidence of relapse are clinical barriers that still call for improvement. Relapse in patients with high-risk, advanced, or refractory disease has limited the overall success of allo-HCT in this setting. Predicting NRM in advance is important as it allows us to determine the potential benefit of allo-HCT. Different scoring systems have been developed with the goal of stratifying a patient’s NRM risk (Table 1). Graft-versus-host disease (GVHD), the most frequent complication after allo-HCT, can cause considerable early and late morbidity, as well as mortality.

### 4.2. Definition of Transplant Eligible

In order to be able to undergo allo-HCT, a patient must have a suitable donor, a good social support system, a good understanding of the procedure and its risks, and a secure financial net (medical insurance coverage). Patients must be well informed, not only about the transplantation process but also about expected or potential post-allo-HCT events, including GVHD and delayed effects, which may arise only years later; in particular, patients should be informed regarding treatment with glucocorticoids that can have severe side effects in older individuals [8].

Older patients are more likely to be unfit, to have significant comorbidities, and poorer performance status. The AML pathophysiology in the elderly also seems to differ, accounting for a higher likelihood of genetic risk for chemotherapy refractoriness and poorer prognosis [14].

Physiologic age, or functional age, in elderly patients evaluates a patient’s physiologic reserve, or conversely, a patient’s vulnerability. It requires a detailed and comprehensive assessment and correlates more accurately with life expectancy when undergoing allo-HCT [84] and a patient’s eligibility for the procedure. It includes frailty assessment [78], functional status, and self-reported function/limitations (Table 1).

A transplant eligible patient is defined as a patient who at least meets the following criteria:Has a suitable donor available;Can tolerate the indicated conditioning chemotherapy regimen;Can tolerate the planned GVHD prophylaxis regimen and GVHD manifestation(s);Will safely tolerate being in an immunosuppressive state and recover from it.

### 4.3. Patient’s Risk Stratification and Assessment of Health Status

One of the most important advancements in the field of allo-HCT was creating safer access to older patients. This was possible not only by decreasing the conditioning intensity, but also by improvements in supportive care and donor selection [85]. Major supportive care improvements include new antibiotics, growth factors, viral reactivation detection, CMV preventive and pre-emptive measures, and better tolerability profiles. This expansion in the supportive care arsenal has allowed older patients with protracted immune reconstitution to effectively overcome infectious complications, GVHD related complications, and disease relapse. More efficient and adequate therapeutic approaches for the management of GVHD have also increased access to allo-HCT for elderly patients.

Unfortunately, only 6% to 8% of patients aged 60 to 80 years receive allo-HCT [15,86]. The extent of comorbidities and other health burdens among the AML populations impacting the benefits from allo-HCT compared to other therapies remains problematic [87]. Comorbidities increase with increasing age, and consequently increase the risk of mortality and morbidity [19,20,88]. An accurate risk stratification to support the identification of elderly patients who will truly derive benefit from undergoing allo-HCT [83,89] is key (Table 1). The increased prevalence of comorbidities, health problems, and decreased performance status leads not only to increased NRM but also to decreased quality of life (QOL) post-allo-HCT. Evaluation methods, such as scoring systems for predicting complications in advance, are necessary for determining the adaptation of allo-HCT and selecting appropriate conditioning regimens. These also help to identify modifiable factors from non-modifiable factors.

### 4.4. Modifiable Factors

Modifiable factors should be optimized prior to undergoing allo-HCT. Examples of modifiable factors: nutritional related factors (vitamin deficiencies including vitamin D), deficient thyroid function, performance status (modifiable to a certain degree), diabetes and hypertension control, and underlying infections. Regarding underlying infections, the active versus potential risk for infection(s) need to be identified and corrected accordingly.

### 4.5. Non-Modifiable Factors

Factors that are considered non-modifiable typically refer to patient-related factors: patient’s age, the presence of comorbid conditions (e.g., smoking history, lung performance, and coronary artery disease), underlying AML-related factors such as remission status [81,82] vs. access and impact of maintenance therapy, and socioeconomic support system. AML-related cytogenetic abnormalities and associated mutation profile (e.g., NGS) have consistently been identified as the strongest determinants of relapse after allo-HCT. This is true regardless of patients’ age or comorbid conditions, although some studies in patients with AML or MDS showed higher relapse rates in older patients when compared to younger patients [8]. To handle the stress associated with allo-HCT, patients need a good social support system and a secure financial net.

The hematopoietic cell transplant comorbidity index (HCT-CI) [19], commonly used as a risk stratification tool for allo-HCT related risks, fails to take into consideration significant underlying geriatric vulnerabilities [79], even when using the age-adjusted HCT-CI [80]. Lin et al. [6], through comprehensive geriatric assessment, identified impairment in instrumental activities of daily living (ADL) to be associated with increased NRM and renal dysfunction, Karnofsky performance status (KPS) [77], cytomegalovirus serostatus, and DRI, all associated with reduced OS.

The Glasgow Prognostic Score (GPS), which assesses the combined C-reactive protein and albumin ratio (CAR), was reported to predict patient survival when they have solid-organ malignancies independently of received chemo/radiotherapy and stages of cancer; this is a prognostic indicator among different inflammatory and nutritional status biomarkers for allo-HCT in elderly patients [89]. Pre-transplant inflammation is a poor prognostic factor in allo-HCT. Pre-transplant CRP is associated with early NRM after allo-HCT [90,91,92] and serum ferritin (SF) is also considered to be an indicator of inflammation [93]. There is compelling evidence that patients with elevated levels of pre-transplant SF have worse prognoses with allo-HCT [91,93,94,95,96,97,98,99].

### 4.6. Conditioning Regimen Selection

The preparative regimen, or conditioning regimen, consists of the administration of chemotherapy with or without total body irradiation (TBI) for the eradication of malignant cells and induction of immune tolerance in the recipient so the infused donor stem cells engraft properly. For patients who are older and have comorbidities and/or have history of extensive chemotherapy prior to undergoing allo-HCT, RIC or NMA conditioning regimens are preferred [100]. The advantages of using RIC include less transfusion requirements due to the transient post-transplant pancytopenia, less chemotherapy-induced liver damage, and less radiation-induced lung damage [101]. The RIC and NMA regimens rely more on the GVL effect, and less on chemotherapy-mediated eradication of leukemia cells. Consequently, the relapse rates after RIC and NMA are higher.

### 4.7. Donor Selection

Less than a third of the patients have an HLA-matched related donor (MRD) or MUD available. Unrelated donor registries worldwide currently include about 30 million volunteer donors; however, most of them are in North America and Europe. The probability of finding a full MUD depends on the patient’s ethnic background and varies on average between 15% and 75% [102,103]. Although allo-HCT from MRD and MUD lead to the best outcomes, they are not always available. In addition, for older patients, if a MRD is available, she/he will be of older age as well, and oftentimes will have significant comorbidities including a history of cancer(s). HLA-haploidentical donors, who share a single HLA haplotype with recipients, are nearly always available and are younger if they are the patient’s children. The percentage of haploidentical allo-HCT with RIC/NMA conditioning intensity has been progressively increasing [104], which suggests that more patients > 60 years have been undergoing allo-HCT. Time to transplant is another factor contributing to OS [105]. The search for an unrelated donor takes, on average, 8 weeks, and not all potential donors listed in the database will be available on a timely basis or even at all. During the past two decades, the emergence and refinement of HLA-haploidentical allo-HCT (haplo-HCT) with post-transplant cyclophosphamide (PTCy) for selective and cytotoxic depletion of alloreactive T-cells while preserving non-alloreactive T-cells [106] has significantly increased access to allo-HCT in general, including for older patients.

In general:Grafts from younger donors (<40 years old) contain more stem cells (CD34+ cells), DFS is higher and relapse is lower for allo-HCT using younger MUDs [107], and younger donors are associated with decreased late mortality and treatment failure [108];MRDs are considered the best option for allo-HCT as HLA matching and shared non-HLA genetic polymorphisms reduce alloimmune reactions and contribute to optimal outcomes due to fast immunological reconstitution and lower incidence of acute GVHD [109,110]. Although the outcomes with MRD and MUD are comparable [111], MRDs are favored because of their faster and more cost-effective workup. However, when a related donor is likely to carry the same genetic mutation as the patient or may be significantly older than unrelated donors, unrelated donors may be preferred;Younger donor age (≤40 years old) has been shown to be a predictor for improved survival in older patients with AML and MDS and receiving PTCy-haplo-HCT [112]. One aspect related to the selection of donor(s) for a haplo-HCT is the importance of the presence of donor-specific antibodies (DSA) in the recipient. They mediate graft rejection in HLA-mismatched allo-HCT [113]. Recipients planning to undergo a haplo-HCT should be screened for DSA, and the donor who carries HLA alleles targeted by these DSA should be avoided. A test for DSA is considered positive when mean fluorescence intensity (MFI) is above 1000, and graft failure risk increases significantly when DSA levels are >5000 MFI at transplant [114]. A thorough review can be found in Timofeeva OA et al. [115].

Investigators worldwide continue to further improve the outcomes of patients treated with PTCy-haplo-HCT, the major goals being the reduction of NRM and GVHD incidence, improving immune reconstitution, and reducing relapse rates. A unique aspect of PTCy-haplo-HCT is the underlying disease relapse marked by loss of the unshared HLA molecules by malignant cells, limiting the use of donor lymphocytes infusion (DLI) as part of a salvage strategy [116]. This unique aspect reinforces the importance of studying and understanding each disease relapse post-allo-HCT as it bears implications on the potential salvage strategies.

### 4.8. Post-Allo-HCT Maintenance Therapy Considerations

Relapse is the principal cause of treatment failure after allo-HCT [117,118]. About 30% of patients with AML relapse within the first 2 years post-allo-HCT [118] and have extremely poor prognosis. While the traditional approach has been to offer induction therapy in AML with the goal of a remission and to proceed to allo-HCT as consolidation, the availability of tolerable drugs increases the appeal of maintenance therapy, particularly for patients with substantial risk for relapse (e.g., pre-transplant detectable MRD).

a.Hypomethylating Agents

While maintenance with HMA after induction therapy for AML appears to improve patients’ outcomes, there is less evidence of its benefit after allo-HCT. A prospective randomized trial of post-transplant azacitidine in 181 patients treated for AML or MDS was randomized for each patient to either receive azacitidine or to serve as a control. No significant difference was noted between RFS and OS. In addition, 23 of the 87 patients randomized to the treatment arm were required to stop the intervention before reaching the study endpoint due to toxicity, patient preference, or for logistical reasons [119]. In other studies with azacitidine, a rather modest benefit regarding prolonged EFS and OS and the suggestion of the maximum tolerated dose at 32 mg/m^2^ for 5 days every 28 days was shown [120,121,122,123].

Single agent decitabine maintenance was shown to be safe, offering some benefits despite toxicities such as myelosuppression and increased infection rates [124,125,126]. A combination of low dose decitabine (15 mg/m^2^ × 3 days) and venetoclax 200 mg (D1 to D21) starting on D+100 was shown to be safe and to decrease relapse rate [127].

CC-486 (oral azacitidine) maintenance offered at different dosing schedules was studied in 30 patients (median age, 64.5 years) post-allo-HCT [128] and in morphologic CR. This was associated with low rates of relapse and GVHD, and a 1-year OS of 86% in the 7-day CC-486 dosing.

b.IDH mutated AML

Enasidenib and ivosidenib, IDH2 and IDH1 inhibitors, respectively, were shown to be safe and well-tolerated as maintenance therapy post-allo-HCT in phase 1 clinical trials [129,130] and with promising results regarding PFS and OS. These studies will be important in establishing the toxicity profile in the posttransplant setting and potentially give an estimate of expected clinical outcomes. The molecular profile of disease at the time of relapse, with particular attention to IDH mutational status, will be critical.

c.FLT3-mutated AML

Both sorafenib [131,132,133] and midostaurin [134] have been studied in the maintenance setting; however, only sorafenib provided a clear benefit to the patients but with significant toxicities.

d.Donor lymphocytes infusion

DLI has the potential to restore GVL immunologic surveillance, driven in part by restoration of T-cell immunity and reversal of T-cell exhaustion in resident CD8+ T cells [135]. However, its efficacy varies depending on the underlying hematologic malignancy. Response rates have been shown to be more modest in AML than in CML, for example [136], and the recommendation is to concomitantly attempt aggressive weaning and discontinuation of immunosuppression upon detection of MRD and/or increasing mixed chimerism, together with close monitoring for GVHD [137,138] signs and symptoms, a known complication.

Preemptive DLI to prevent impending relapse, upon detection of MRD or mixed chimerism, is another strategy. Despite encouraging results, the evidence comes mostly from small single-center retrospective studies [139,140,141].

Prophylactic DLI is administered to patients at high risk for relapse posttransplant, at a defined interval, to prevent disease recurrence [142]. The administration of prophylactic azacitidine in combination with up to three DLIs at escalating doses was studied in 30 patients with high-risk AML or MDS. A decrease in relapse rate to 27.6% vs. 41.9% in control patients was reported, although it was not significant (*p* = 0.21) [143].

**CASE 4**—A 62-year-old female was admitted with pancytopenia. Her WBC count was 3200/µL, absolute neutrophil count 600/µL, hemoglobin 6 g/dL, and platelet count 89,000/µL. Peripheral blood flow cytometry showed 5% myeloblasts, CD34, CD117, and CD33 positive, and CD14, CD13, and TdT negative, which is immunophenotypically consistent with AML. She underwent induction therapy with liposomal cytarabine plus daunorubicin (CPX-351, Vyxeos^®^, Palo Alto, CA, USA).

### 4.9. Case 4 Overview

Her disease carried a complex karyotype (including multiple chromosomes losses such as -4, -5, -7, -8, and -18) and NGS showed *P53* -R248Q mutation, VAF 52%. The patient’s age and *P53* mutational status indicate a very poor prognosis [9]. In high-risk AML, CPX-351 demonstrated improved CR, OS, and QoL, and a higher proportion of patients suitable for allo-HCT with superior OS post allografting when compared to patients receiving standard “7 + 3” [144,145,146,147,148]. However, CPX-351 provided no benefit over “7 + 3” to patients with *P53*-mutated AML [149,150], which is present in 10%–20% of patients [151], and even with allo-HCT, patients experienced poor OS [152,153]. As the most effective approach for the management of TP53-mutated AML remains unclear, clinical trial discussions should be prioritized.

### 4.10. Recent Advances

CPX-351 (Vyxeos^®^, daunorubicin and cytarabine—liposome, Jazz Pharmaceuticals, Palo Alto, CA, USA), is a liposomal encapsulation of cytarabine and daunorubicin in a synergistic 5:1 molar ratio [154]. Compared to the conventional “7 + 3” regimen for older adults with high-risk AML, CPX-351 is associated with improved remission rates, EFS, OS, and OS without allotransplant [144,145,150]. There were also higher rates of allo-HCT and OS after allo-HCT in fit adults with high-risk AML, defined as therapy-related (tAML), secondary (sAML) with a history of MDS or chronic myelomonocytic leukemia (CMML), or de novo AML in older adults [145,146,147,148]. Improved efficacy of CPX-351 may be due to liposomal-related pharmacokinetics that prolong plasma half-life and enhanced uptake by leukemic cells by bypassing chemo-resistant mechanisms and maintaining the synergistic ratio within leukemic cells. Additionally, CPX-351 has similar adverse effects to “7 + 3”, with the exception of further prolonged thrombocytopenia and neutropenia by approximately one week [144] and a similar 60-day related mortality; however, it has a superior improving QOL [146]. Interestingly, lengths of stay were two weeks shorter, and more than one third of tAML or sAML patients received induction CPX-351 as outpatients [155]. Since its approval, encouraging results have been reported when administered with venetoclax [156] or FLT3 inhibitors [157]. There are several clinical trials open for de novo and R/R AML evaluating different combinations. The V-FAST study [158] evaluated the safety and preliminary efficacy of CPX-351 plus midostaurin in 232 patients, showing a manageable safety profile and encouraging remission rates (82% CR after the first induction cycle) in de novo *FLT3^mut^* AML. The rate of myelotoxicity was comparable to single agent CPX-351, and 56% and 33% of patients carrying *FLT3-ITD* and *FLT3-TKD* mutations, respectively, proceeded to allo-HCT.

### 4.11. Role of Maintenance Therapy in Transplant-Ineligible Elderly Patients with AML

For transplant-ineligible elderly patients with AML, maintenance therapy may offer persistent CR and prolonged DFS through less intensive therapy. Multiple early studies evaluating maintenance azacitidine or decitabine in transplant-ineligible AML patients treated with intensive therapy failed to show DFS or OS benefit [159,160,161,162]. More recently, the QUAZAR AML-001 study evaluated 472 transplant-ineligible AML patients post intensive induction chemotherapy randomized to receive oral azacitidine maintenance vs. placebo until progression [163]. Patients receiving maintenance azacitidine had significantly prolonged mOS when compared to the placebo group (24.7 vs. 14.8 months, *p* < 0.001), and importantly, a preplanned subgroup analysis showed a consistent OS for patients >65 years [163]. Although the benefit for maintenance azacitidine in this patient population is well established, whether maintenance HMA plus venetoclax would also be of benefit remains unclear. Early results from a single-arm phase II trial showed the efficacy and safety of maintenance azacitidine plus venetoclax in 33 patients with AML who were not immediately eligible for allo-HCT, after intensive or low-intensity induction, with a 1-year OS of 93.8% post intensive induction and 53.3% post low-intensity induction, associated with 18% incidence of grade 3/4 infection [164]. Longer duration of follow-up and larger randomized trials comparing different doublet maintenance strategies are necessary to evaluate their survival benefit over single agent HMA in the maintenance setting.

Thus far, molecularly targeted agents such as FLT3 inhibitors (sorafenib or quizartinib) have been approved in the maintenance setting post-allo-HCT, but not as maintenance therapy in patients who are transplant-ineligible [55,133]. Early phase I data demonstrated the safety and efficacy of ivosidenib and enasidenib with intensive chemotherapy and as maintenance therapy in newly diagnosed IDH1/IDH2-mutated AML, but these agents are not FDA approved in the maintenance setting [48]. A phase III clinical trial is currently ongoing to evaluate the clinical efficacy of ivosidenib and enasidenib with intensive induction and as maintenance therapy for patients with IDH1/IDH2-mutated AML (NCT03839771).

## 5. Future Directions and Discussion

Is Liposomal cytarabine plus daunorubicin superior to other induction regimens in transplant eligible patients? MRC UK NCRI AML-19 trial sub-cohort analysis.

To date, most CPX-351 clinical trials have been performed in the older AML population. The UK NCRI AML-19 trial examined the efficacy of the liposomal compound against conventional FLAG-IDA (fludarabine, high-dose cytarabine, idarubicin, and G-CSF) in younger adult patients (median age 56 years, range 18–70) [165]. Myelodysplasia-related chromosomal abnormalities and complex karyotypes were observed in 78% and 49% of these patients, respectively. *P53, FLT3 ITD* and *ASXL* were found in 44%, 19%, and 18% of cases, respectively. Interestingly, induction response, EFS, and OS were similar in both groups. However, a subgroup analysis demonstrated superior efficacy among AML patients with detectable myelodysplasia-related genetic mutations (38.4 vs. 16.3 months in CPX-351 vs. FLAG-IDA, *p* = 0.05), rather than clinically or cytogenetically defined AML-MRC [165].

### 5.1. Where Are We with Oral Agent Inductions?

Six out of the eight new agents for AML introduced since 2017 have predominantly been oral agents. The benefits of oral agents include decreased length of hospital stay, lower risk of infection, and better QoL for patients. Wide implementation of an “oral only” AML-directed therapy retains important challenges. A careful patient selection to estimate the risk of (a) TLS, (b) disseminated intravascular coagulation, and (c) differentiation syndrome (DS) should be emphasized. Presently, only *IDH1/IDH2*-mutated AML benefit from single agent “oral only” induction alternatives, and in the pivotal ivosidenib trial, patients with performance status > 2 were ineligible [44]. Serious adverse events, including differentiation syndrome, occurred in 79% of patients, although investigators were able to manage them through weekly follow-ups. Also, studies are needed to investigate the efficacy of triplet therapy in *IDH1*/*IDH2*-mutated AML disease patients.

The idea of “chemotherapy free” interventions are very appealing. Bazinet et al. evaluated the efficacy and safety of ASTX727 (oral decitabine/cedazuridine) plus venetoclax in patients with AML in the frontline and R/R settings [166], median age 79 (range 50–92) and 71 years (range 46–75) in the frontline and R/R groups, respectively. Among 52 patients, 14%, 7%, and 79% were favorable, intermediate, and adverse risk by ELN-2022. ORR was 67% (CR + CRi = 62%) and 50% (CR + CRi = 50%) in frontline and R/R patients. OS was 12.8 and 7.6 months, respectively. The most common side effects included neutropenic fever and sepsis in 26% and 8% of patients, respectively, and dose reductions were necessary in 67% of cases during cycle 2. It is probable that additional combinations, including triplet oral therapies, will significantly advance the therapeutic AML landscape.

### 5.2. What Are the Most Promising Oral Agents Recently Approved and in Development for AML?

#### 5.2.1. Newer IDH Inhibitors

Olutasidenib (Olu), a potent and selective *IDH1* inhibitor, has been recently approved by the FDA for treatment of R/R *IDH1*^mut^ AML. The ORR was 48% (CR + CRh = 35%), with a median duration of response of 25.9 months [167]. Interestingly, among 12 patients with prior BCL-2 inhibitor exposure, ORR was 50% and DS was observed in 7% of patients. In the 4 arms olutasidenib and azacytidine (Aza) combination trial (Olu-Aza), the CR + CRh rates were 45%, 47%, 38%, and 30% in treatment naïve, R/R disease without HMA or IDH exposure, prior HMA exposure, and prior IDH inhibitor therapy, respectively [168]. Interestingly, in a recent phase 1/2 multicenter study, the post-allo-HCT cohort (31 patients) achieved a CR + CRi rate of 29% and an ORR of 32%, with 50% of the responders having received single agent Olu. Duration of response was 7.1 months (range 1–23.4 months), allowing 3 patients to receive a second allo-HCT [169].

#### 5.2.2. Newer Oral HMA+ Cedazuridine

After the success encountered with the development and approval of oral decitabine/cedazuridine (DEC-C, Inqovi^®^, Otsuka Pharmaceutical Co., Pleasanton, CA, USA) for MDS and CMML, preliminary animal data of the oral DNA methyltransferase inhibitor (DNMTi) azacitidine (AZA) in combination with cedazuridine suggests a comparable pharmacokinetic to intravenous AZA [170,171]. Phase 2 clinical trials are currently accruing in the United States evaluating the clinical efficacy in MDS, CMML, and AML.

### 5.3. Where Are We with Potentially Stopping HMA Plus Venetoclax in Responder Subgroups?

A retrospective study evaluated 29 AML patients (median age 74, range 65–80) treated with venetoclax plus either low dose cytarabine or HMA, and in remission for a minimum of 12 months, comparing their outcomes when continuing (55% patients) or discontinuing therapy (45% patients) [172]. A significantly improved OS in those who “had stopped” therapy (median 71.3 months vs. 43.7 months) was noted, with predictors of “stop” success associated with *NPM1* and *IDH2* mutations and MRD-negative status at the time of therapy cessation. Larger prospective randomized clinical trials are needed prior to the generalization of these provocative findings.

### 5.4. How to Improve Outcomes in Elderly AML Patients Receiving Allo-HCT

In general, fit elderly patients, who are motivated and active and with less comorbidities are the patients who derive the most benefit from allo-HCT. However, further studies are needed to determine the following:The optimal approach to performing haplo-HCT and PTCy;Impact of early incorporation of multidisciplinary care, including attention to physical therapy, nutrition, and psychosocial health that may improve a patient’s fitness to undergo and tolerate allo-HCT in general;How to best improve GVHD prophylaxis and treatments available.

Survivorship programs are integral to enhancing long-term outcomes and QoL of transplant recipients in general, and all efforts are needed to increase access to meaningful therapies that increase QoL and GVHD-free survival (GFRS) in post-allo-HCT patients.

However, our patients still encounter significant shortfalls in their AML managements through allo-HCT: safer (acceptable NRM) and improved outcomes for older patients with significant comorbidities, further decrease in disease relapse risks, effective and tolerable maintenance therapies including pre-emptive DLI, safe and optimized early tapering of immunosuppression, and meaningful monitoring of MRD/early relapse.

## 6. Conclusions

Despite significant therapeutic progress, the survival of elderly patients with AML remains unacceptably dismal, especially among intermediate and adverse ELN-2022 risk groups. A better understanding of the human genome aging process, inducing epigenetic, transcriptomic, and immunological modifications, will support novel and more performant trial designs. Currently, therapy selection should be based on how well elderly patients are able to tolerate intense interventions (i.e., liposomal cytarabine plus daunorubicin) versus hypomethylating agents plus venetoclax. Allocation to induction intensity is mostly based on performance status. However, the pathogenic and prognostic role of MDS-related mutations in regimen selection are of major interest given the correlation for response and MDS ontogeny. Meanwhile, a promising number of AML-directed interventions, including cellular therapies, are expected to improve the survival of this vulnerable population. Finally, concerning allo-HCT, an area of interest is the mitigation of the “age acceleration” effect, significantly promoted by patients’ comorbid conditions including organ dysfunctions, with the goal of improving elderly AML patients’ outcomes from allo-HCT.

## Figures and Tables

**Figure 1 biomedicines-12-00975-f001:**
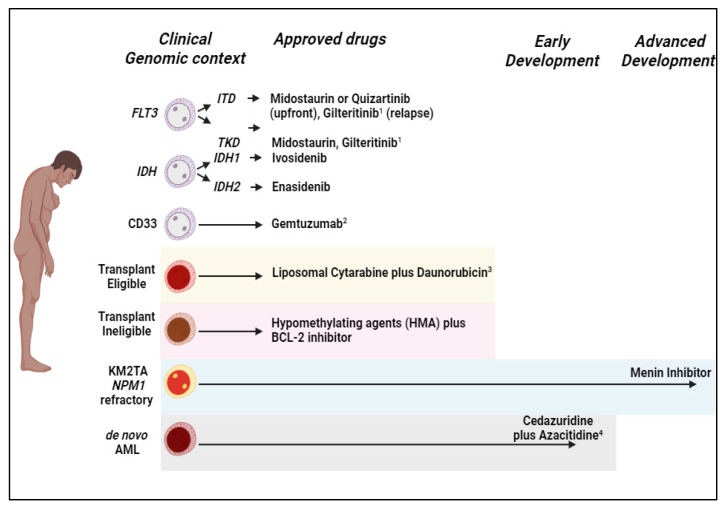
Approved and in development therapies for elderly patients with acute myeloid leukemia. ^1^ Approved for R/R AML only; ^2^ Anti-CD33 monoclonal antibody approved for therapy of core binding factor (CBF) AML in combination with intravenous cytarabine plus anthracycline (i.e., idarubicin or daunorubicin); ^3^ Liposomal cytarabine plus daunorubicin (CPX-351) approved for myelodysplasia-related and therapy-related AML; ^4^ Phase 2 trial oral cedazuridine plus azacitidine in subjects with myelodysplastic syndrome (MDS), chronic myelomonocytic leukemia (CMML), or acute myeloid leukemia.

**Table 1 biomedicines-12-00975-t001:** Patient Risk Assessment Prior to Allogeneic HCT.

	Risk Stratification Model	Variable(s)	Stratification	Comments
**Patient-related**	**Charlson Comorbidity Index (CCI)** [5]	Number of comorbid conditions	Scores = 1-year OS 0 = 12% 1–2 = 26% 3–4 = 52% 5 or >5 = 85%	Primarily developed for patients admitted to general medical ward; lacks prior infections and psychiatric disturbances that have a bearing on HCT outcomes. Excellent inter-rater reliability, predicts long-term mortality in different clinical population. Specific ICD coding needed for proper allocation of scoring.
	**Karnofsky Performance Status** [77]	Daily activity level, ability to perform ordinary tasks	Scores range from 0 to 100, where higher score means that patient is better able to carry out activities.	Global health status, reliable predictor of NRM and OS after transplant. Extremely easy to perform but is subjective; supplement by frailty measure.
	**Fried’s Frailty Phenotype (FFP)** [78]	(a) gait speed (b) grip strength (c) activity level (d) exhaustion (e) weight loss	≥3 criteria = frail 1–2 criteria = pre-frail 0 criterion = fit	Age and FFP associated with restricted mean survival time. Trials for pre-HCT interventions to reverse frailty and incorporation of AML therapy type are needed. Requires objective assessments for accurate phenotype capture.
	**Geriatric Assessment (GA)** [79]	(a) Functional status, evaluated by ECOG performance (b) Frailty, by Fried frailty index (FI) (c) Comorbidity, by HCT-CI (d) Mental health * (e) Nutritional status, Alb < 3.5, self-reported weight loss (f) Degree of inflammation, determined by serum CRP >10 mg/L	Scores = 2-year OS 0 = 62% 1 = 44% 2 = 13%	203 patients ≥50 years, median 58 (range 50–73) Limitations in instrumental ADLs, slow walk speed, high HCT-CI, low mental health, and elevated CRP were significantly associated with inferior OS. May support creation of transplant supportive care package targeting GA-defined limitations. However, study also includes younger patients (50 to 65 year old) and does not discriminate based on prior treatment modalities.
**Transplant-related**	**Hematopoietic Cell Transplantation Specific Comorbidity Index (HCT-CI)** [19]	(a) Refined comorbidity definitions (b) Evaluate increased severity of comorbidities in correlation with toxicity risk and mortality	3 risk groups: low risk (score 0) vs. intermediate (score 1–2) vs. high (score ≥ 3) score 0: 2y-OS 71%, 2y-NRM 14%; score 1–2: 2y-OS 60%, 2y-NRM 34%; score ≥3: 2y-OS 34%, 2y-NRM 41%	Refinement in comorbidities definition, introduction of lab and functional testing criteria allowing accurate assessment and replicability across independent observers. Only model prospectively validated in two large studies: higher level of evidence. Online tool: http://www.hctci.org/ (accessed on 15 March 2024).
	**Age-adjusted HCT-CI** [80]	Age is integrated	Age ≥ 40 years is assigned a weight of 1 Added to the HCT-CI to constitute a composite comorbidity/age index	Age is a poor prognostic factor, less applicable to elderly patients with AML as all of them are >40 years
**Disease-related**	**European Leukemia Network (ELN) 2022** [9]	*Favorable * Core binding factor (CBF): t (8;21); inv16 or t (16;16); bZIP CEBPA NPM1 without FLT3-ITD	Transplant in CR2 or if persistent MRD	Includes the following: revised genetic risk classification, revised response criteria, and treatment recommendations.
		*Intermediate * Wild type NPM1 with FLT3 ITD Mutated NPM1 with FLT3 ITD t (9;11) (p21.3; q23.3)/MLLT3::KM2TA Cytogenetic aberrations not considered favorable or adverse	Transplant in CR2 or if persistent MRD	Includes management of special situations (hyperleukocytosis, leukostasis), DIC, TLS, DS, and supportive care (anti-infectious prophylaxis and transfusions).
		*Unfavorable * U2AF1, SF3B1, SRSF2, STAG2, RUNX1, ASXL1, P53, and complex karyotype	Transplant in CR1	
	**Disease Risk Index (DRI)** [81]	*Disease type * (AML vs. ALL vs. CML vs. MDS vs. MPN vs. DLBCL vs. T-cell lymphoma) *Remission status * CR1 or CR2 vs. PR vs. induction failure vs. active disease)	4 groups: low vs. intermediate vs. high vs. very high	Stratification by disease and disease status at HCT Not restricted to AML Applicable across different cytogenetics grouping Similar outcomes for MAC and RIC groups are noted
	**Refined DRI** [82]	Includes additional entities: (a) ALL Philadelphia+ and Philadelphia− (b) MDS classified based on blast%, cytogenetic, and response to therapy (c) Burkit lymphoma (BL) (d) Mantle cell lymphoma (MCL)	4 groups: low vs. intermediate vs. high vs. very high 2-year OS ranging from 64% to 24%	Cohort of >13,000 patients Conditioning intensity–independent index Does not include molecular information
**Combined**	**European Group for BMT Risk Score** [83]	Risk factors: (a) Age (b) Disease stage (c) Time interval and diagnosis to transplant (mo) (d) Donor type (e) Donor–recipient sex combination	Seven groups are defined, with different related TRM and OS Age: <20, 20–40, >40 Disease stage: early, intermediate, and late Time interval: <12 mo and >12 mo Donor: HLA-identical sibling vs. unrelated donor Donor–recipient sex combination: all other vs. D:F, R:M	Incorporates time interval from diagnosis and in AML Can have discordant impact—longer time from CR1 is associated with decreased relapse.

Abbreviations: ALL, acute lymphoblastic leukemia; AML, acute myeloid leukemia; CML, chronic myeloid leukemia; CR, complete remission; CRP, C-reactive protein; DLBCL, diffuse large B-cell lymphoma; MDS, myelodysplastic syndrome; mo, months; MPN, myeloproliferative neoplasm; MRD, measurable residual disease; NRM, non-relapse mortality; OS, overall survival; ADL, activities of daily living (includes help with bathing, dressing, walking, eating, grooming, taking medications, and shopping). * evaluated from Mental Component Summary of the Medical Outcomes Short Form-36 health-related quality of life questionnaire (SF36-MCS).

## Data Availability

Data available through PubMed and other publicly available resources as stated within the manuscript.

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
