# Peer review of "Transplant Eligible and Ineligible Elderly Patients with AML—A Genomic Approach and Next Generation Questions"

_biomedicines, 2024, doi:10.3390/biomedicines12050975_

Round 1
Reviewer 1 Report
Comments and Suggestions for Authors
The review entitled: “Transplant Eligible and Ineligible Elderly Patients with AML- A Genomic Approach and Next Generation Questions” by Sackstein et al. addressed in their review challenging aspects related to age, predisposition to hematopoietic malignancies and current modifications by the ELN-2022 risk stratifications underlined by four clinical cases to highlight the management of elderly patients with AML undergoing allo-HCT.
The review is well written and of special interest for the readership.
Minor comments:
Comments:
1. Section “Transplant considerations”: line 315-316: “…still carry a significant risk of NRM”. The authors should also add in this sentence the additional risk of relapse (see also page 9 line 415-416) due to the RIC and NMA.
2. Within the section of non-modifiable factors (or better patient-related factors) of (elderly) patients, the authors should highlight more intensively the advantages and disadvantages of each tool to assess patient-related risk factors e.g. within the table.
3. Moreover, the authors should also discuss more intensively potential allocation to therapy in elderly patients due to patient-related risk factors by current tools.
Author Response
- Section “Transplant considerations”: line 315-316: “…still carry a significant risk of NRM”. The authors should also add in this sentence the additional risk of relapse (see also page 9 line 415-416) due to the RIC and NMA.
Dear Reviewer - This portion requested was added.
2. Within the section of non-modifiable factors (or better patient-related factors) of (elderly) patients, the authors should highlight more intensively the advantages and disadvantages of each tool to assess patient-related risk factors e.g. within the table.
These were added in table.
3. Moreover, the authors should also discuss more intensively potential allocation to therapy in elderly patients due to patient-related risk factors by current tools.
These were modified and added in table.
Gustavo Rivero MD

Reviewer 2 Report
Comments and Suggestions for Authors
In the current manuscript, the authors comprehensively summarize the current therapeutic options for acute myeloid leukemia (AML) based on the risk assessment genomic findings. The manuscript is well summarized, and this is so helpful for the readers to understand the therapeutic options. Here are my comments that would improve the already great manuscript.
1. I think the case reports in the current manuscript are so helpful for the readers to understand the current status and future challenges. However, I'm wondering how the authors selected the cases. If these cases were designed by the authors for the current manuscript, it would be better to describe it in the introduction.
2. What does the image of human mean in Figure 1?
3. As the authors described clonal hematopoiesis, are there any studies on the influence of clonal hematopoiesis for the donor selection?
Author Response
- I think the case reports in the current manuscript are so helpful for the readers to understand the current status and future challenges. However, I'm wondering how the authors selected the cases. If these cases were designed by the authors for the current manuscript, it would be better to describe it in the introduction.
Dear Reviewer: all cases are real from our leukemia unit at Georgetown University.
2. What does the image of human mean in Figure 1?
It illustrates an elderly AML patient.
3. As the authors described clonal hematopoiesis, are there any studies on the influence of clonal hematopoiesis for the donor selection?
Dear Reviewer - To date, no major studies have been performed that would allow donor selection based on evidence or not of pathogenic myeloid variants (within donor graft).
